# Male-Biased Adult Production of the Striped Fruit Fly, *Zeugodacus scutellata*, by Feeding dsRNA Specific to *Transformer-2*

**DOI:** 10.3390/insects11040211

**Published:** 2020-03-28

**Authors:** Md. Abdullah Al Baki, Mohammad Vatanparast, Yonggyun Kim

**Affiliations:** Department of Plant Medicals, Andong National University, Andong 36729, Korea; albaki121@gmail.com (M.A.A.B.); mvatanparast@yahoo.com (M.V.)

**Keywords:** transformer-2, SIT, RNA interference, dsRNA, *Zeugodacus scutellata*

## Abstract

Sterile insect release technique (SIT) is effective for eradicating quarantine insects including various tephritid fruit flies. When SIT is used for fruit flies, it is challenging to remove females from sterile males due to oviposition-associated piercing damage. This study developed a sex transition technique by feeding double-stranded RNA (dsRNA) specific to a sex-determining gene, *Transformer-2* (*Zs-Tra2*) of the striped fruit fly, *Zeugodacus scutellata*. *Zs-Tra2* is homologous to other fruit fly orthologs. It is highly expressed in female adults. RNA interference (RNAi) of *Zs-Tra2* by injecting or feeding its specific dsRNA to larvae significantly increased male ratio. Recombinant *Escherichia coli* cells expressing dsRNA specific to *Zs-Tra2* were prepared and used to feed larvae to suppress *Zs-Tra2* gene expression levels. When these recombinant bacteria were fed to larvae during the entire feeding stage, the test population was significantly male-biased. Some females treated with such recombinant *E. coli* exhibited mosaic morphological characters such as the presence of male-specific abdominal setae in females. This study proposes a novel technique by feeding dsRNA specific to *Transformer-2* to reduce female production during mass-rearing of tephritid males for SIT.

## 1. Introduction

A number of tephritid fruit flies are designated as quarantine pests in many countries [1]. These fruit flies are classified into the Tephritoidea superfamily of about 7300 species [2] with two possible origins based on molecular phylogenetic analysis [3]. In Tephritoidea, the largest family is Tephritidae, which possesses more than 4400 species [4,5]. In Korea, 90 tephritid species have been identified, of which two species (*Zeugodacus depressa* and *Z. scutellata*) are regarded as insect pests [6].

The striped fruit fly, *Z. scutellata*, occurs in Asian countries such as Korea, Japan, Taiwan, China, Bhutan, Thailand, India, and Malaysia [7]. Its outbreak can cause more than 70% damage to pumpkin flowers, thus posing serious threat to crop production [8]. Chinese cucumber (*Trichosanthes kirilowii*) is the main wild host of *Z. scutellata.* Its alternative hosts include wild gourds (*Diplocyclos palmatus*, *Zehneria liukiuensis*, and *Trichosanthes* spp.) [9,10,11]. Although chemical and biological control tactics have been developed to control fruit fly populations, sterile insect release technique (SIT) is known to be effective in eradicating the fruit flies [12]. 

SIT is a genetic control method of specific target insect species using a massive release of sterile males to compete with wild males in mating with females. In addition, females mated with sterile males will produce no offspring [13]. In a SIT program, one obstacle is sex sorting after mass-rearing to release only males [14]. A genetic sexing line using chromosomal translocating technology was generated to facilitate the sorting of females by exhibiting different colored pupae of the melon fly, *Z. cucurbitae*, in which brown pupae are males while white pupae are females [15]. For Mediterranean fruit flies, *Ceratitis capitata*, a transgenic approach has been used to develop an effective genetic sexing line with females killed at the early embryonic stage by adding transgenic apoptotic genes controlled by female-specific gene expression [16]. However, classical genetic or transgenic sexing systems have practical limitations, such as the extra labor needed for sorting pupae or genetically modified organism issue. To overcome these obstacles, a novel molecular approach focusing on manipulation of the sex determination system of insects is needed. 

Like sex determination systems known in other animals, insect sex is determined by sex chromosomes. However, to be differentiated into males and females, chromosomal information is interpreted by sex-determining genes of *doublesex* (*Dsx*) for expressing sex-specific morphological characters and *fruitless* (*Fru*) for expressing sex-specific behavior. These genes have been well characterized in *Drosophila* [17]. Briefly, X-linked signal at XX females can activate *Sex lethal* (*Sxl*) which is a sex determination master gene. Subsequent functional Sxl protein product can help make a functional *transformer* (*Tra*) protein by preventing alternative splicing generating premature stop codon. The functional Tra protein can bind to *transformer-2* (*Tra-2*), an RNA-binding protein. This complex allows female-specific alternative splicing of *Dsx* mRNA. In contrast, males without an X-linked signal cannot form a functional Sxl protein. Thus, they will fail to make functional Tra proteins, leading to no Tra/Tra-2 protein complex and male-specific alternative splicing of *Dsx* mRNA. Tephritids such as *C. capitate* have been well studied. They do not have the Sxl. However, *Dominant Male Determiner* (*M*) on the Y chromosome is presumed to inhibit Tra protein which in turn prevents the formation of female-specific *Tra* mRNA [18]. Tra-2 is required for female development because the RNA interference (RNAi) of *Tra-2* expression in XX females can lead to phenotypic males of *C. capitata* [19]. When this RNAi technique against *Tra-2* expression was applied to eggs of the Oriental fruit fly, *Bactrocera dorsalis*, it produced almost 100% males [20]. Similar approach has been applied to the pumpkin fruit fly (*Z. tau*), resulting in male-biased sex ratio and some intersexes [21]. However, egg injections of double-strand RNA are a practical limitation to produce males only in mass-rearing for SIT. 

To overcome such a practical obstacle of RNAi strategy, this study used a feeding technique for dsRNA against *Tra-2* identified from *Z. scutellata* (*Zs-Tra2*). To analyze its function, RNAi was performed on young larvae by injecting dsRNA specific to *Zs-Tra2*. To be practical, dsRNA was fed to *Z. scutellata* to determine its RNAi efficiency. To continuously and economically feed dsRNA, a recombinant *Escherichia coli* expressing dsRNA specific to *Zs-Tra2* was constructed and supplied to feed for larvae. This study provides a novel technology for the production of male-biased fruit flies for SIT.

## 2. Materials and Methods

### 2.1. Insect Rearing and Morphological Characters for Sex Determination

*Z. scutellata* larvae were collected from a pumpkin field in Andong, Korea and reared on a larval artificial diet consisting of 1 g agar, 5 g pumpkin powder, 5 g yeast, 4 g wheat bran, 5 g sugar, 4 g casein, 1 g citric acid, 0.5 g sodium benzoate, 0.1 g methyl-p-hydroxybenzoate, and 0.02 g streptomycin in 73.38 mL water. Laboratory conditions were: temperature of 26 °C, relative humidity of 70–75%, and photoperiod of 14:10 h (L:D). Under these laboratory conditions, larvae underwent three instars (L1–L3) for approximately 4.5 days. Fresh pumpkin stems and flowers were provided in adult cages for oviposition. Adults were fed an adult artificial diet (2 g sucrose, 1 g yeast, and 1 g casein). Laid eggs were collected and transferred to larval artificial diet. For pupation, a larval growth box was provided with sand. Pupae were kept in the box under laboratory conditions until adults emerged. Female and male characters of adults were externally determined with oviposition and abdominal setae, respectively (Appendix A). Internally, ovary and testis were observed to clarify the sex.

### 2.2. Sequence Analysis

Tra-2 sequence alignment and similarity analysis was performed using the BLAST program on the NCBI website (http://www.ncbi.nlm.nih.gov). Phylogenetic tree of Tra-2 was constructed using MEGA6 program. Bootstrapping values were obtained with 1000 repetitions to support branch and gene clusters. Pfam (http://pfam.xfam.org) was used to predict protein domain structure. Exon and intron region of Tra-2 were predicted using the Spidey program (http://www.ncbi.nlm.nih.gov/spidey/).

### 2.3. RNA Extraction and cDNA Preparation

Total RNAs were extracted from all developmental stages (30 eggs, 10 young larvae (L1–L2), 5 L3 larvae, one pupa, or one adult) using Trizol reagent (Invitrogen, Carlsbad, CA, USA) according to the manufacturer’s instruction. Extracted RNA was treated with DNase to remove genomic DNA (gDNA) contamination and confirmed no gDNA by no PCR product using the extracted RNA as template. After RNA extraction, RNA was resuspended in nuclease-free water and quantified using a spectrophotometer (NanoDrop, Thermo Scientific, Wilmington, DE, USA). Extracted RNA (1 µg) was then used for cDNA synthesis with RT PreMix (Intron Biotechnology, Seoul, Korea) containing oligo dT primer according to the manufacturer’s instruction.

### 2.4. RT-PCR and RT-qPCR

RT-PCR was conducted using DNA Taq polymerase (GeneALL, Seoul, Korea) with an initial heat treatment at 94 °C for 2 min followed by 35 cycles of DNA denaturation at 94 °C for 1 min, primer annealing at 50 °C, extension at 72 °C for 1 min. A final chain extension step was performed at 72 °C for 10 min. All gene expression levels in this study were determined using a real-time PCR machine (Step One Plus Real-Time PCR System, Applied Biosystem, Singapore) under the guideline of Bustin et al. [22]. RT-qPCR was conducted with a reaction volume of 20 µL containing 10 µL of Power SYBR Green PCR Master Mix (Thermo Scientific Korea), 5 µL of cDNA template (50 ng), 1 µL of forward primer, and 1 µL of reverse primer (Appendix A). After 10 min of an initial denaturation step, qPCR was performed with 40 cycles of denaturation at 94 °C for 1 min, annealing at 50 °C for 1 min, and extension at 72 °C for 1 min. Fluorescence emitted from the newly synthesized PCR products at every cycle was monitored to quantify PCR products. Melting curves of PCR products were analyzed to confirm single product. Expression level of actin as reference was used to normalize target gene expression level after different treatments. Quantitative analysis was performed using the comparative CT (2^−ΔΔCT^) method [23].

### 2.5. Preparation of Double Stranded RNA (dsRNA) with In Vitro Transcription

Template DNA was amplified with gene-specific primers (Appendix A) containing T7 promoter sequence (5′-TAATACGACTCACTATAGGGAGA-3′) at their 5′ ends. dsRNA was prepared using a Megascript RNAi Kit (Ambion, Austin, TX, USA) according to the manufacturer’s instruction. The resulting PCR product was used to in vitro to synthesize dsRNA specific to *Zs-Tra2* using T7 RNA polymerase with NTP mixture at 37 °C for 4 h. The synthesized dsRNA was purified with a filter cartridge (10051G2, Thermo Fisher Scientific, Vilnius, Lithuania: 10 kDa cut-off size) and resuspended with deionized water.

### 2.6. Injection or Feeding Treatment of dsRNA to Young Larvae

For the dsRNA injection, 0.3 µL (500 ng/μL) of purified dsRNA was injected to early L1 (<12 h) larval hemocoel with a microinjector (PV830, World Precision Instruments, Sarasota, FL, USA). Treated larvae were then provided with larval artificial diet ad libitum and reared under laboratory conditions until pupation. As dsRNA control (dsCON), dsRNA specific to a viral gene, *CpBV-ORF302*, was prepared using the method of Park and Kim [24]. 

For feeding dsRNA, 10 µL (500 ng/μL) of purified dsRNA was mixed with 250 mg of artificial diet. The treated diet was then used to feed five larvae at early L1. At 48 h after feeding, treated larvae were fed with untreated fresh artificial diet and maintained until pupation. The control larvae were fed diet mixed with dsCON.

### 2.7. Preparation of dsRNA with Recombinant Bacteria

L4440 vector was kindly provided by Seung Jae Lee (Pohang University of Science and Technology, Pohang, Korea). Multiple cloning sites were flanked at both ends with T7 promoters in an inverted orientation. PCR primers used in in vitro dsRNA preparation were used to amplify DNA fragments followed by digestion with restriction enzymes of *Hind* III and *Xba* I. Digested DNA fragments were ligated to the L4440 vector. A recombinant vector was used to transform competent cells of *Escherichia coli* HT115 (DE3) lacking RNase III by electroporation. Postive clones were selected on ampicillin Luria–Bertani (LB) medium containing 100 μg/mL ampicillin (AMP) and confirmed by sequencing the insert fragment.

To produce dsRNA from recombinant bacteria, a single colony was picked and inoculated to AMP-LB medium followed by incubation at 37 °C with shaking (250 rpm) for 16 h. To the cultured broth (5 mL), 500 mL of fresh LB medium containing 100 μg/mL ampicillin was then added followed by incubation at 37 °C until bacteria reached their exponential growth phase with OD_600_ = 0.6~0.7. Expression of T7 polymerase was induced by adding 0.4 mM (final concentration) of IPTG. Bacteria were then incubated at 37 °C for 4 h. These cultured cells were harvested by centrifugation at 7000× *g* for 30 min.

### 2.8. Quantification of dsRNA Produced from Recombinant Bacteria

Total bacterial RNA was extracted using RNA extraction mini kit (Qiagen Korea, Seoul, Korea). The dsRNA was confirmed by electrophoresis on 1% agarose gel. Quantification of dsRNA amounts of recombinant bacteria followed the method of Kim et al. [25]. A standard curve was generated with known amounts of purified dsRNA synthesized by in vitro transcription described above. RNA band intensity was quantified with an image analyzer (Image Lab Software, Bio-Rad Korea, Seoul, Korea). 

### 2.9. Pretreatment of Recombinant Bacteria Using Heat and Sonication

After overexpression and re-suspending bacterial cells in distilled water, two sequential pre-treatments were applied before feeding larvae. Bacterial cells were killed by heat treatment at 95 °C for 10 min. Bacterial cell walls were then disrupted with an ultrasonicator (Bandelin Sonoplus, Berlin, Germany) at 95% intensity with 10 cycles of 10 min burst. Bacterial viability was assessed by plating 100 μL of treated bacterial sample on LB plate containing ampicillin (100 μg/mL) to confirm no bacterial growth after heat treatment.

### 2.10. Feeding Assay of Recombinant Bacteria Expressing dsRNA

Recombinant bacteria (10 µL, 6.5 × 10 ^8^ cells/mL) were mixed with 50 mg of larval artificial diet. Treated diet was used to feed five larvae at early L1. Every 48 h, the diet was replaced with a freshly treated diet. As a control, non-recombinant HT115 bacteria were used to treat the diet. Each treatment was replicated three times. For each replication, 100 larvae were used.

### 2.11. Statistical Analysis

All studies were performed with three independent replications. Results are presented as mean ± standard deviation and plotted with Sigma plot (Systat Software, San Jose, CA, USA). Means were compared by least square difference (LSD) test of one-way analysis of variance (ANOVA) using SAS program [26]. Significance was discriminated at Type I error = 0.05.

## 3. Results

### 3.1. Morphological Characters of Female and Male Z. scutellata Adults

Female adults had significantly (*p* < 0.05) longer body length than males (Appendix A) even when the ovipositor was not included. Both female and male *Z. scutellata* adults had six visible abdominal segments. Males had 12–18 long pleural setae on the third abdominal segment (Appendix A). Reproductive organs such as ovary and testis were recognized just after adult emergence (Appendix A). In females, a pair of ovaries were connected to oviducts ending with ovipositor while a pair of testes were connected to vas deferens which were linked to the ejaculatory duct along with the accessory gland in males. 

### 3.2. Prediction of Transformer-2 (Zs-Tra2) of Z. Scutellata and Expression Profile

*Zs-Tra2* sequence was obtained from GenBank with accession number of MT103663. Zs-Tra2 protein was predicted to possess signature motifs such as arginine/serine-rich domain (RS), ribonucleoprotein region (RNP), RNA recognition motif (RRM), and a linker region (LR) between RRM and RS (Figure 1A). Phylogenetic analysis indicated that *Zs-Tra2* was clustered with other dipteran Tra-2 genes, in which it shared the highest homology (88%) with that of *B. dorsalis* (Figure 1B). Expression patterns of *Zs-Tra2* in different developmental stages of *Z. scutellata* were analyzed (Figure 2). *Zs-Tra2* was expressed in all developmental stage from eggs to adults with high expression levels in females. In females, it was expressed in all three body parts, such as head, thorax, and abdomen (data not shown).

### 3.3. Male-Biased Sex Ratio after RNAi of Zs-Tra2 Expression

To determine the function of *Zs-Tra2* in sex determination, *Zs-Tra2* expression was silenced by RNAi (Figure 3). To silence *Zs-Tra2* expression, its specific dsRNA was injected to first instar larvae which significantly (*p* < 0.05) reduced its mRNA level at 24 h after injection. The microinjection gave little injection damage (<1%) on the larval survival. The RNAi effect was maintained for at least 72 h after injection (Figure 3A). Under this condition, RNAi-treated larvae had 63% of males while control larvae had only 41% males (Figure 3B). 

To determine whether RNAi effect could be achieved by oral administration, dsRNA was delivered via feeding after mixing dsRNA with the larval diet (Figure 4A). L1 larvae were treated for 24 h with a diet containing dsRNA. The *Zs-Tra2* expression level was significantly (*p* < 0.05) decreased for two days after the feeding treatment. However, it then increased back to the level of untreated larvae. RNAi feeding also resulted in a male-biased sex ratio (Figure 4B).

### 3.4. Construction of Recombinant E. coli Expressing dsRNA Specific to Zs-Tra2 and Its Effect on Sex Transformation

To express dsRNA in *E. coli*, a partial (218 bp) sequence of *Zs-Tra2* was inserted between opposite T7 promoters (Figure 5A). The recombinant L4440 expression vector was then used to transform *E. coli* and the dsRNA was over-expressed by adding IPTG inducer (Figure 5B). Extracted RNA from recombinant bacteria contained dsRNA specific to *Zs-Tra2*. The amount of dsRNA produced from such recombinant *E. coli* was linearly proportional to bacterial density (Figure 5C).

### 3.5. Effect of Feeding Recombinant E. coli on Sex Ratio of Z. scutellata

Feeding recombinant bacteria significantly (*p* < 0.05) reduced mRNA levels of *Zs-Tra2* (Figure 6A). This RNAi effect resulted in a male-biased sex ratio in adults (Figure 6B). Compared to the control, RNAi-treated flies had more males by over 30%. Interestingly, some females (containing ovipositor) in the RNAi-treated group developed male-characteristic abdominal setae (Figure 6C). The intersex-like individuals had the ovary. 

## 4. Discussion

This study developed a novel sexing system by feeding dsRNA against *Z. scutellata*, a serious insect pest of pumpkin crops [8]. Its larvae feed pumpkin flowers within stamen which prevents their exposure to chemical insecticides [27]. Although a mixture treatment of male lure (raspberry ketone or cuelure) with insecticides has been effective for controlling its population [28], SIT has been regarded as one of the most effective control tactics against this insect pest [29]. An obstacle for SIT is the tedious sorting out of females to be removed. To develop an effective sexing system, a comprehensive understanding of sexual differentiation in *Z. scutellata* offers an opportunity to promote the development of novel sexing strategies and genetic control techniques to fight this pest. In this regard, the altering sex determination system was exploited by RNAi in this study.

High expression levels of *Zs-Tra2* were required for female sex differentiation in *Z. scutellata*. *Zs-Tra2* was expressed in all developmental stages. It was also expressed in both male and female adults. However, it was expressed in much higher levels in females than in males. Tra-2 protein and RNA-binding protein 1 are associated with Tra protein to form a splicing regulatory complex that mediates the splicing of its targets, *Dsx* and *Fru* mRNAs in *Drosophila melanogaster* [30]. Tra-2 belongs to the family of RNA binding proteins containing an RNA-recognition motif (‘RRM’) domain which plays crucial role in regulating mRNA splicing of *Dsx* mRNA into a female type variant [31,32]. Without Tra-2, *Dsx* mRNA is spliced into a male-specific Dsx transcript variant. In *D. melanogaster*, two mRNA variants of *Tra-2* are expressed in somatic tissues of both sexes. The other two variants are present only in male germline [33]. Somatic Tra-2 products mediate the alternative splicing of *Dsx* mRNA while two male Tra-2 proteins are functioning in spermatogenesis [34,35,36]. In tephritid fruit flies, mRNA variants of *Tra-2* have been reported. For example, three mRNA variants of *Tra-2* have been identified in *Z. tau*, of which at least one is expressed in both sexes and the other is specific to the embryo or germline [21]. Although *Zs-Tra2* mRNA variants were not assessed in our current study, its expression profile in males suggested the presence of variants of the *Zs-Tra2* transcript.

The expression of RNAi specific to *Zs-Tra2* resulted in a male-biased sex ratio in *Z. scutellata*. A microinjection of dsRNA to L1 larvae was effective in increasing its male production. Feeding dsRNA to growing larvae also increased its male ratio. In various fruit flies other than *Drosophila*, Tra-2 genes have been identified. They play crucial roles in the regulation of female-specific splicing of *Dsx* pre-mRNA. These include 12 *Anastrepha* species [37,38], *Bactrocera oleae* [39], *B. tryoni*, *B. jarvisi* [40], *B. dorsalis* [20], *B. tau* [21], and *C. capitata* [19]. The knocking down of *Tra-2* expression in these tephritid fruit flies can interrupt the splicing pattern of *Dsx* pre-mRNA of females to differentiate females into male phenotypes [41,42]. Compared to these previous studies using a dsRNA injection in an early embryo, our current result of RNAi was obtained by injection or the feeding of dsRNA to growing larvae. Injecting dsRNA into an early embryo is likely to be more effective in producing males than injecting or feeding dsRNA to young larvae. In *B. dorsalis*, nearly 100% males were obtained with embryonic RNAi against Tra-2 [20]. In *Z. tau*, dsRNA injection to embryo has produced 72.4% males while the control has produced only 38.6% males [21]. This indicated an increase of 187.6% in male production. In contrast, dsRNA injections into larvae produced only a 153.7% increase in male production compared to control injection. The feeding of dsRNA resulted in much less efficiency (125.0% increase). The lower efficiency of feeding dsRNA might be explained by a lower amount of dsRNA targeting mRNA compared to injecting dsRNA because of the environmental RNAi. In general, the efficiency of RNAi depends on the amount of exogenous dsRNA application [43]. Thus, injecting dsRNA into embryos is more effective than dsRNA treatment against larvae. However, injecting dsRNA to young embryos or larvae is not practical for mass-rearing to perform SIT. Thus, feeding dsRNA to larvae would be ideal to increase male sex ratio.

Recombinant *E. coli* was constructed to produce dsRNA specific to *Zs-Tra2*. The amount of dsRNA produced was proportional to the bacterial cell number. Based on the regression equation, the feeding treatment with recombinant bacteria used 12.5 μg of dsRNA in a 250 mg diet. This dose of dsRNA was 2.5 times higher than feeding dsRNA prepared by in vitro transcription. The injection of dsRNA into young larvae used only 15 ng dsRNA per larva. This can be explained by the difficulty in the release of dsRNA from the bacteria in the insect gut. To facilitate dsRNA release from bacteria, this study pre-treated bacteria with heat or ultra-sonication to disrupt the bacterial cell walls and membranes. While there was no significant difference in insecticidal activity between live and heat-killed bacteria, a pretreatment of sonication can significantly enhance the ability to kill *Spodoptera exigua* insects [25]. The degradation of dsRNA in the gut lumen likely occurs by dsRNase attacks originating from the midgut, as seen in several insects including *Bombyx mori* [43] and *Lygus lineolaris* [44] or from salivary secretion during feeding in a sucking insect, *Acyrthosiphon pisum* [45]. Thus, the actual number of dsRNA molecules released from the transformed *E. coli* might be remarkably reduced due to the limitation of dsRNA release and the degrading factor in the gut of *Z. scutellata*. The sex transformation technique by feeding dsRNA was supported by the development of intersex individuals detected in some females. This might be explained by the mosaic expression of male- or female-type *Dsx* expression by fluctuating *Zs-Tra2* mRNA levels among different tissues. These results indicated that the dsRNA feeding technique was effective in suppressing Zs-Tra2 expression to alter sex ratio biased to males. In particular, recombinant bacteria expressing the dsRNA are effective and economical for a mass rearing system to perform SIT. Furthermore, more than 80% of RNAi-generated males could mate with normal females in a study on *B. dorsalis* [20], suggesting that feeding dsRNA is an ideal approach in a genetic sexing system to achieve male-biased production in SIT.

## 5. Conclusions

This study aimed to develop a technique for the sex transformation of *Z. scutellata* using sex determination system. A certain level of *Zs-Tra2* expression was required for female sex differentiation. Its expression level was suppressed by RNAi specific to *Zs-Tra2* by feeding dsRNA during the larval period. Feeding recombinant *E. coli* expressing dsRNA was also effective in suppressing *Zs-Tra2* expression and resulted in a male-biased sex ratio. This study introduces a novel technique for sex transformation by feeding dsRNA, which would be applicable for the mass rearing of quarantine fruit flies to be eradicated by SIT.

## Figures and Tables

**Figure 1 insects-11-00211-f001:**
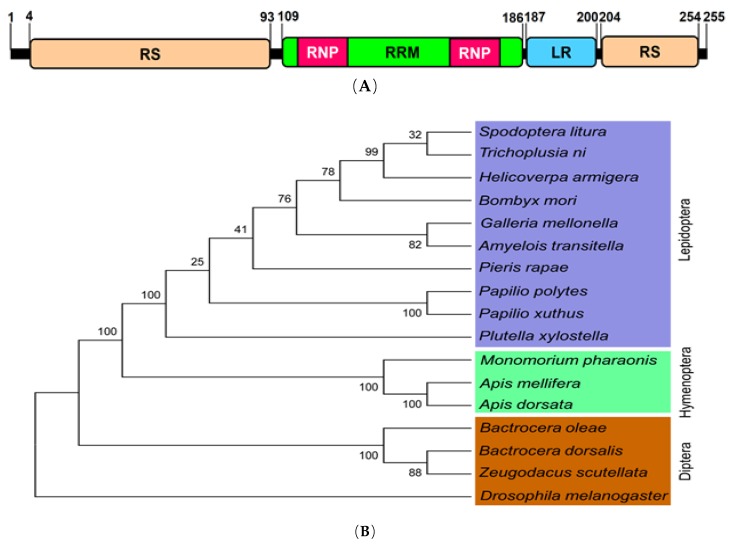
Molecular characters of Transformer-2 gene (*Zs-Tra2*) of *Zeugodacus scutellata*. (**A**) Functional domains and motifs were predicted with Pfam (https://www.pfam.xfam.org/): Arginine/serine rich domain (‘RS’), RNA recognition motif (‘RRM’), ribonucleoprotein region (‘RNP’), and linker region (LR). (**B**) Phylogenetic analysis of *Zs-Tra2* with other insect orthologs. Amino acid sequences of *Transformer-2* were retrieved from GenBank with accession numbers of XP_012553366.1 for *Bombyx mori*, NP_001252514.1 for *Apis mellifera*, XP_021199210.1 for *Helicoverpa armigera*, XP_022838015.1 for *Spodoptera litura*, XP_006623037.1 for *Apis dorsata*, XP_026762058.1 for *Galleria mellonella*, XP_012539181.1 for *Monomorium pharaonis*, CAD67988.1 for *Bactrocera oleae*, ALJ94044.1 for *Bactrocera dorsalis*, XP_013197999.1 for *Amyelois transitella*, XP_013145601.1 for *Papilio polytes*, XP_013170155.1 for *Papilio xuthus*, XP_011560377.1 for *Plutella xylostella*, XP_022120857.1 for *Pieris rapae*, XP_026724961.1 for *Trichoplusia ni*, and AAF58232.2 for *Drosophila melanogaster*. Amino acid sequences were aligned with MEGA6 program. Bootstrapping values were obtained with 1000 repetitions to support branch and clustering.

**Figure 2 insects-11-00211-f002:**
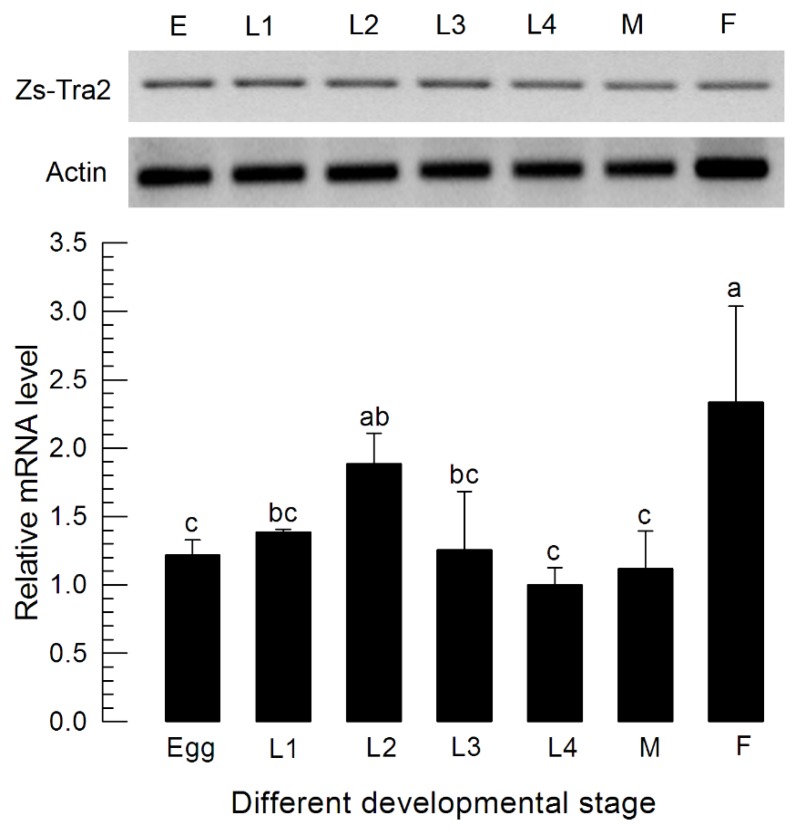
Expression profile of *Zs-Tra2* in different developmental stages including different larval instars (‘L1–L3’), male (‘M’) adults, and female (‘F’) adults. Actin gene was used as a reference gene to normalize target gene expression level in RT-qPCR. All treatments were independently replicated three times. Different letters above standard deviation bars indicate significant difference among means at Type I error = 0.05 (LSD test).

**Figure 3 insects-11-00211-f003:**
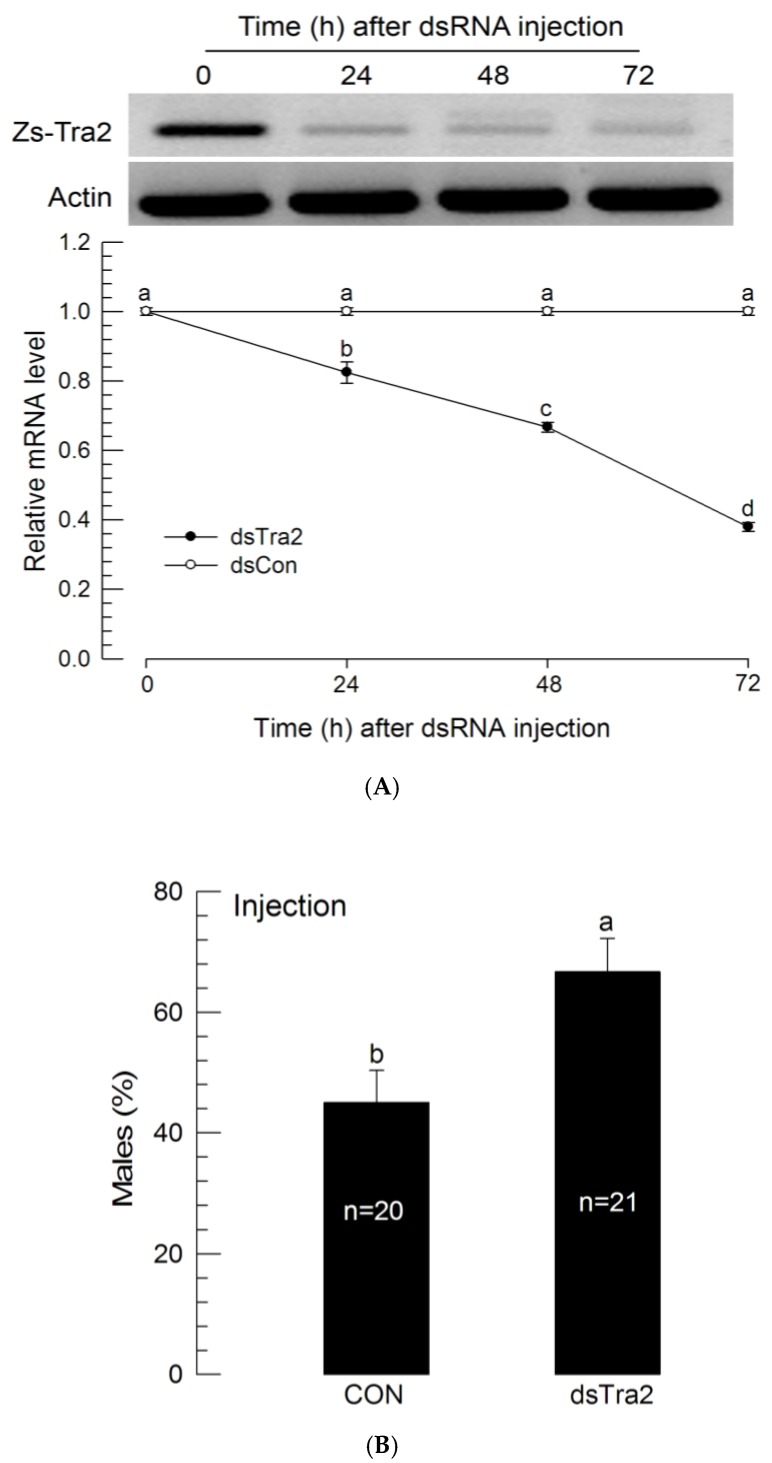
Influence of dsRNA (‘dsTra2’) specific to *Zs-Tra2* on male-biased adult production in *Z. scutellata*. RNAi was performed by injecting 0.03 μL (500 ng/μL) of dsRNA to newly molted L1 larvae (<12 h) by microinjection. For dsRNA control (‘dsCon’), dsRNA specific to CpBV-ORF302 was prepared with the method of Park and Kim (2010). (**A**) RNAi efficiency was assessed by RT-qPCR. Actin was used as a reference gene of RT-qPCR to normalize target gene expression level. Each treatment was replicated with independently prepared samples. (**B**) Outcome was assessed in ratio of male adults. Number indicate the number of assessed adults. Different letters above standard deviation bars indicate significant difference among means at Type I error = 0.05 (LSD test).

**Figure 4 insects-11-00211-f004:**
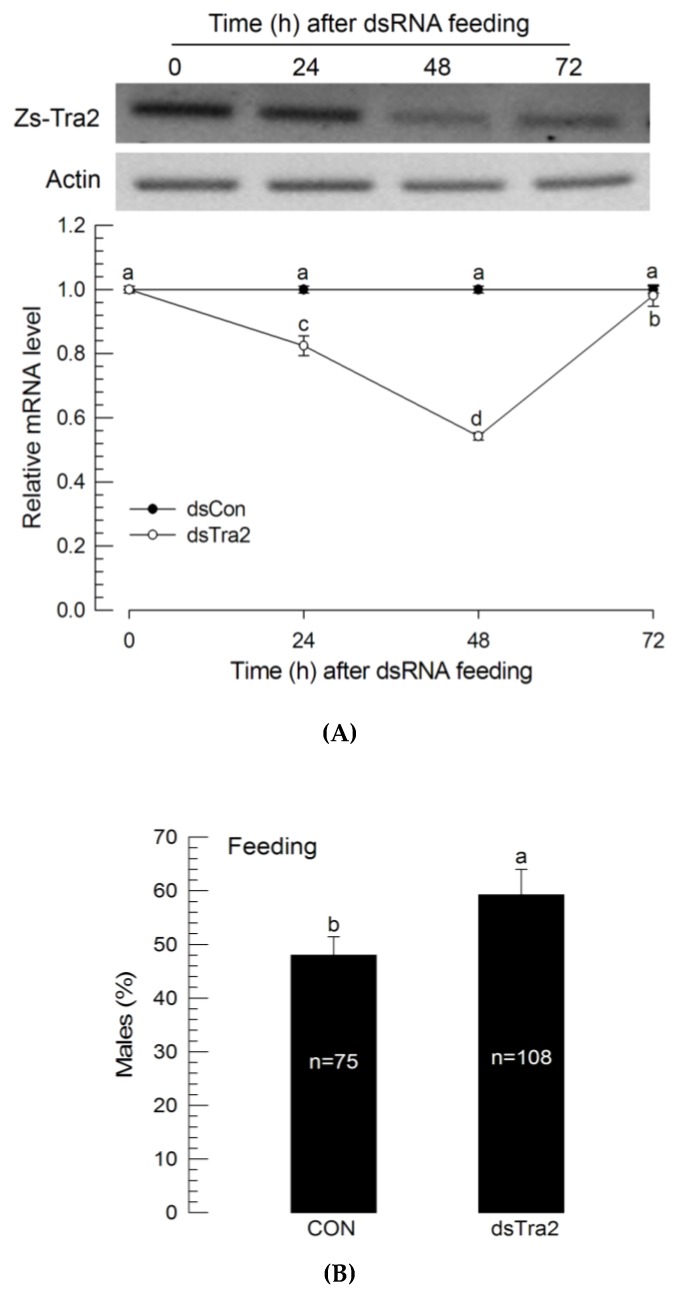
Influence of dsRNA (‘dsTra2’) specific to *Zs-Tra2* on male-biased adult production in *Z. scutellata*. RNAi was performed by feeding dsRNA (10 μg) mixed with diet (500 mg) to newly molted L1 larvae (<12 h) for 48 h. For dsRNA control (‘dsCon’), dsRNA specific to CpBV-ORF302 was prepared by the method of Park and Kim [24]. (**A**) RNAi efficiency assessed by RT-qPCR. Actin gene expression was used as a reference gene of RT-qPCR to normalize target gene expression level. Each treatment was replicated with independently prepared samples. (**B**) Outcome was assessed in ratio of male adults. Number indicates the number of assessed adults. Different letters above standard deviation bars indicate significant difference among means at Type I error = 0.05 (LSD test).

**Figure 5 insects-11-00211-f005:**
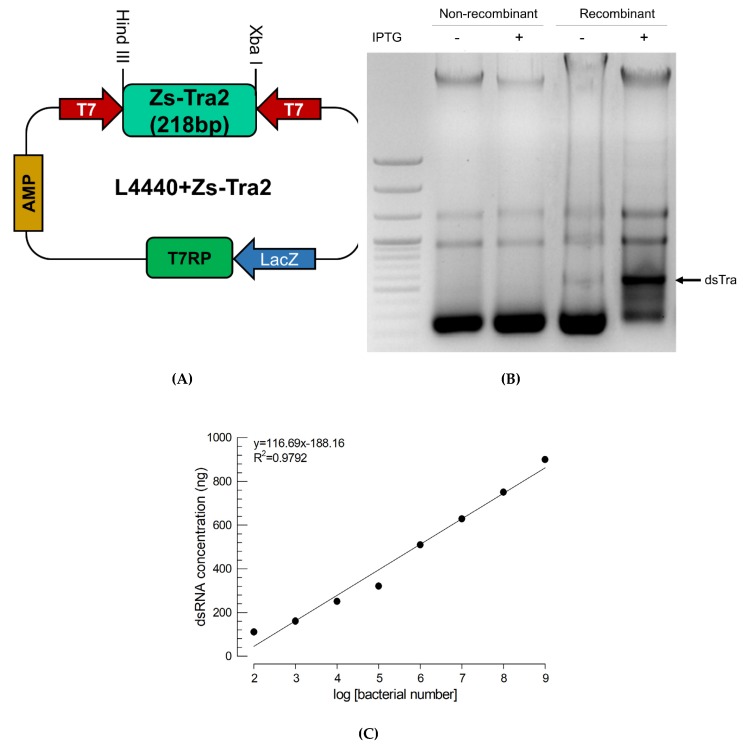
Recombinant *E. coli* expressing dsRNA (*dsTra2*) specific to *Zs-Tra2*. (**A**) Cloning of a fragment of *Zs-Tra2* into an expression vector of L4440 using *Hind* III and *Xba* I multiple cloning sites. After selecting transformed cells using ampicillin (‘AMP’) medium, T7 RNA polymerase (‘T7 RP’) was overexpressed by IPTG inducer on lactose promoter (‘LacZ’). Partial *Zs-Tra2* RNA is transcribed by two opposite T7 promoters (‘T7’). (**B**) Confirmation of dsTra2 production specifically in recombinant *E. coli* under IPTG induction. (**C**) Regression analysis of dsTra2 amounts from bacterial cell numbers. The extracted dsRNA was quantified by band intensity based on a standard curve using known amounts of dsRNA prepared by in vitro construction method. Each point represents an average obtained from three replications.

**Figure 6 insects-11-00211-f006:**
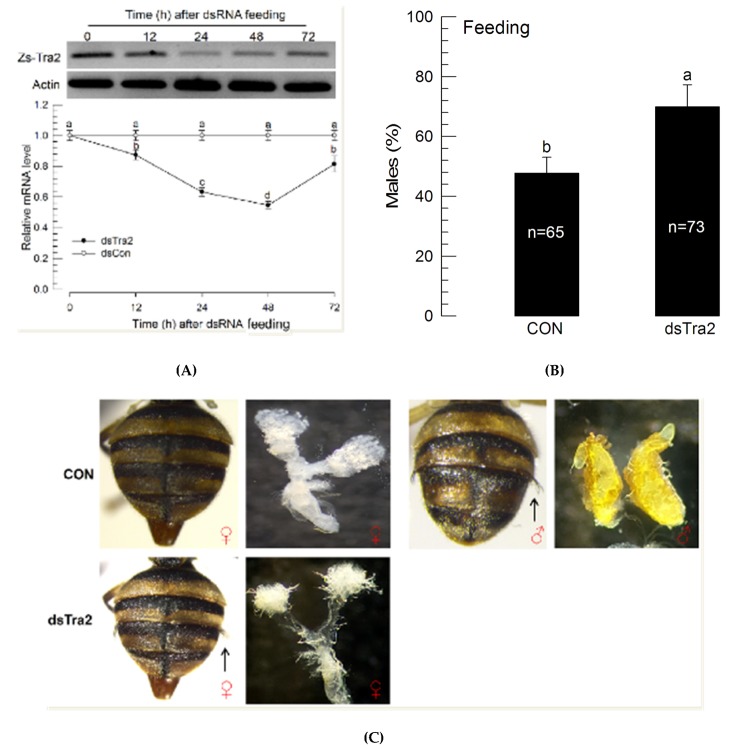
Influence of recombinant *E. coli* expressing dsRNA (‘dsTra2’) specific to *Zs-Tra2* on male-biased adult production in *Z. scutellata*. RNAi was performed for newly molted L1 larvae (<12 h) by feeding diet (500 mg) containing recombinant *E. coli*, producing 25 μg of dsTra2. Control (‘dsCON’) used non-recombinant bacteria. (**A**) RNAi efficiency assessed by RT-qPCR. Actin was used as a reference gene of RT-qPCR to normalize target gene expression level. Each treatment was replicated with independently prepared samples. (**B**) Outcome assessed in ratio of male adults. Around 300~400 newly hatched L1 larvae were used for both control and treatment. Number indicates the number of assessed adults. Different letters above standard deviation bars indicate significant difference among means at Type I error = 0.05 (LSD test). (**C**) Phenotypic analysis in abdominal structures including ovipositor for female character and abdominal pleural setae (see arrows) for male character.

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
