# Peer review of "Male-biased Adult Production of the Striped Fruit Fly, Zeugodacus scutellata, by Feeding dsRNA Specific to Transformer-2"

_insects, 2020, doi:10.3390/insects11040211_

Round 1

Reviewer 1 Report

This study reports on the manipulation of the sex-determining gene of a tephritid fly, Zeugodacus scutellata with the aim to bias sex towards male for use in mass rearing facilities.

Overall the manuscript is well written, the methods are appropriate, the data is robust and aptly discussed.

A few minor comments should be addressed.

Grammar. A number of sentences are missing definite and indefinite articles.

Methods

Line 128 Need to explain what the dsRNA control (CpBV-ORF302) is and why selected.

Results

Figures: For clarity the positions of A, B and C to identify different component of figures would be better placed at the top of each individual figure rather than the bottom.

Section 3.1. Morphological characteristics

There is no corresponding section in the methods for morphological characteristics, and it is unclear, until section 3.4 and figure 7 why this was included.

Figures 4 and 5.

When dsTra2 is injected, relative levels of mRNA decrease over a 72 hour period, but when fed levels recover between 48 and 72 hours. Is there an explanation for this?

Figure 7A. X-axis label should be time after dsRNA feeding, not injection?

Figure 7C.  Perhaps an arrow could be included to point to the abdominal setae on dsTra2 females and males. What is the purpose of including photos of the reproductive organs as these are not mentioned in the text or figure legends.

The significance of abdominal setae occurring in females should be discussed.

Discussion

Line 306. You state that In tephritid fruit flies, mRNA variants of Tra-2 have been reported, and although not assessed in this study, can you comment on whether dsTra2 affects all these variants?

Author Response

Comment #1-1: This study reports on the manipulation of the sex-determining gene of a tephritid fly, Zeugodacus scutellata with the aim to bias sex towards male for use in mass rearing facilities. Overall the manuscript is well written, the methods are appropriate, the data is robust and aptly discussed. A few minor comments should be addressed. A number of sentences are missing definite and indefinite articles.

Response: After all revision according to reviewers' comments, the corrrespnding author read again and added articles to the appropriate words.

Comment #1-2: Line 128 Need to explain what the dsRNA control (CpBV-ORF302) is and why selected.

Response: Added as follows: “As dsRNA control (dsCON), dsRNA specific to a viral gene, CpBV-ORF302, was prepared using the method of Park and Kim [24].”

Comment #1-3: Figures: For clarity the positions of A, B and C to identify different component of figures would be better placed at the top of each individual figure rather than the bottom.

Response: I agree, but the journal keeps the bottom position.

Comment #1-4: Section 3.1. Morphological characteristics. There is no corresponding section in the methods for morphological characteristics, and it is unclear, until section 3.4 and figure 7 why this was included.

Response: The method is now added to the Materials and methods as follows: “Female and male characters of adults were externally determined with oviposition and abdominal setae, respectively (Figure S1). Internally, ovary and testis were observed to clarify the sex.”

Comment #1-5: Figures 4 and 5. When dsTra2 is injected, relative levels of mRNA decrease over a 72 hour period, but when fed levels recover between 48 and 72 hours. Is there an explanation for this?

Response: We add following explanation to discussion: “Less efficiency of feeding dsRNA might be explained by less amount of dsRNA targeting mRNA compared to injecting dsRNA because the environmental RNAi, in general, the efficiency of RNAi depends on the amount of exogenous dsRNA application [44].”

Comment #1-6: Figure 7A. X-axis label should be time after dsRNA feeding, not injection?

Response: Corrected

Comment #1-7: Figure 7C.  Perhaps an arrow could be included to point to the abdominal setae on dsTra2 females and males. What is the purpose of including photos of the reproductive organs as these are not mentioned in the text or figure legends.

Response: Arrows are added to the photo. Explanation on the internal organs are added as follows: “The intersex-like individuals had ovary.”

Comment #1-8: The significance of abdominal setae occurring in females should be discussed.

Response: Additional explanation added as follows: “Interestingly, some females (containing ovipositor) in the RNAi-treated group developed male-characteristic abdominal setae (Figure 6C). The intersex-like individuals had the ovary.”

Comment #1-9: Line 306. You state that In tephritid fruit flies, mRNA variants of Tra-2 have been reported, and although not assessed in this study, can you comment on whether dsTra2 affects all these variants?

Response: It is a great comment and challenging to us. We commented the possibility of Zs-Tra2 transcript variants. However, at this time, we cannot give any evidence to tell the efficacy of dsTra2 on silencing other transcripts.

Reviewer 2 Report

Review of Manuscript ID: insects-749425 entitled “Male-based adult production…specific to Transformer-2

General Comments:

While this is an interesting study on the effects and utility of feeding dsRNA to larvae of the striped fruit fly Zeugodacus scutellate, there are several concerns on the manner in which the results have been presented, the general write-up as well as details of methods which dampen enthusiasm for recommending acceptance of this manuscript in its present form. One important fact that the authors have not addressed is the role of Zs-Tra2 other than in determining sex. Thus, interfering with Zs-Tra2 may also result in effects that may not necessarily be related to male biased adult production. This needs to be fully explored, and in such case mere silencing will not be enough, a CRISPR knock-out is recommended. There are some additional concerns which would require to be addressed as follows:

Specific Comments:

  1. Abstract Line 11-13: Is there any evidence for this claim? If so, it should be dealt in the Introduction and Discussion.
  2. Abstract Line 16: The authors state that Zs-Tra2 is highly expressed in adult females. It would be necessary to show tissue specific expression pattern.
  3. Materials and Methods: What is the average duration in days for development from egg to adult under the conditions specified for scutellata? This needs to be mentioned.
  4. RNA Extraction: How was the integrity of the extracted RNA analyzed? How was gDNA removed?
  5. RT-PCR and RT-qPCR: What was the efficiency of the primers?
  6. dsRNA preparation: What was the cut-off of the filter cartridge used?
  7. Injection or feeding of dsRNA: the % success has to be mentioned.
  8. How were the transformants / recombinant bacteria expressing dsRNA screened?
  9. Did pre-treatment or sonication of bacteria not destroy the dsRNA? Again, how was the integrity of the dsRNA determined?
  10. Results: The morphological characteristics of male and female scutellata can be moved to supplemental data.
  11. Figure 3: While the histogram shows that females have higher Zs-Tra2 RNA, the image does not reflect this. In fact, in the image, the actin gene expression levels appear higher in females than at other stages. This data also depicts that Zs-Tra2 may have other functions which is why it is present in all stages including in males albeit at a lower level than in females.
  12. Figure 4: dsCon should also be depicted in the same gel.
  13. While injection of dsRNA resulted in decreased expression levels of Zs-Tra2 from 24 to 72 hrs (again, the gel does not reflect such change, but only the graph does), feeding resulted in a reversion of Zs-Tra2. A proper explanation of this is required.
  14. The effects as depicted are fairly mild.
  15. Line 270-271. What % females developed male-characteristics?
  16. Why did the authors not try to make the normal bacterial gut flora in Scutellata (choosing any one predominant bacteria) express the dsRNA?

Author Response

Comment #2-1: While this is an interesting study on the effects and utility of feeding dsRNA to larvae of the striped fruit fly Zeugodacus scutellate, there are several concerns on the manner in which the results have been presented, the general write-up as well as details of methods which dampen enthusiasm for recommending acceptance of this manuscript in its present form. One important fact that the authors have not addressed is the role of Zs-Tra2 other than in determining sex. Thus, interfering with Zs-Tra2 may also result in effects that may not necessarily be related to male biased adult production. This needs to be fully explored, and in such case mere silencing will not be enough, a CRISPR knock-out is recommended. There are some additional concerns which would require to be addressed as follows:

Response: We agree on the high efficiency of CRISPR techbique to see the clear influence of Tra-2 on sexing and other developmental processes. However, this study provides an idea to apply dsRNA technique to SIT by reducing female contaminants.

Comment #2-2: Abstract Line 11-13: Is there any evidence for this claim? If so, it should be dealt in the Introduction and Discussion.

Response: We agree on the concern of the reviewer and deleted the sentence.

Comment #2-3: Abstract Line 16: The authors state that Zs-Tra2 is highly expressed in adult females. It would be necessary to show tissue specific expression pattern.

Response: Our preliminary assay showed that it was expressed in females in different body parts. We add this information to text as follows: “In females, it was expressed in all three body parts such as head, thorax and abdomen (data not shown).”

Comment #2-4: Materials and Methods: What is the average duration in days for development from egg to adult under the conditions specified for scutellata? This needs to be mentioned.

Response: The data is added to text as follows: “Under these laboratory conditions, larvae underwent three instars (L1-L3) for approximately 4.5 days. ”

Comment #2-5: RNA Extraction: How was the integrity of the extracted RNA analyzed? How was gDNA removed?

Response: Extracted RNA was treated with DNase and confirmed no gDNA by PCR. This information is added to the M&M as follows: “Extracted RNA was treated with DNase to remove genomic DNA (gDNA) contamination and confirmed no gDNA by no PCR product using the extracted RNA as template. ”

Comment #2-6: RT-PCR and RT-qPCR: What was the efficiency of the primers?

Response: We confirmed the primers by single and expected size of PCR products. For qPCR, we assessed as follows: “Fluorescence emitted from the newly synthesized PCR products at every cycle was monitored to quantify PCR products. Melting curves of PCR products were analyzed to confirm single product. ”

Comment #2-7: dsRNA preparation: What was the cut-off of the filter cartridge used?

Response: We add the information as follows: “a filter cartridge (10051G2, Thermo Fisher Scientific, Vilnius, Lithuania: 10 kDa cut-off size)”

Comment #2-8: Injection or feeding of dsRNA: the % success has to be mentioned.

Response: We added following information to the text: “To silence Zs-Tra2 expression, its specific dsRNA was injected to 1st instar larvae which significantly (P < 0.05) reduced its mRNA level at 24 h after injection. The microinjection gave little injection damage (< 1%) on the larval survival. ”

Comment #2-9: How were the transformants / recombinant bacteria expressing dsRNA screened?

Response: The details are added to the text as follows: “Postive clones were selected on ampicillin Luria-Bertani (LB) medium containing 100 g/mL ampicillin (AMP) and confirmed by sequencing the insert fragment.”

Comment #2-10: Did pre-treatment or sonication of bacteria not destroy the dsRNA? Again, how was the integrity of the dsRNA determined?

Response: We did not check the physical structure of dsRNA after sonication. Instead, we analyzed the dsRNA from bacteria by running gel as shown in Figure 6B.

Comment #2-11: Results: The morphological characteristics of male and female  scutellata can be moved to supplemental data.

Response: Figure 1 is now moved to Figure S1.

Comment #2-12: Figure 3: While the histogram shows that females have higher Zs-Tra2 RNA, the image does not reflect this. In fact, in the image, the actin gene expression levels appear higher in females than at other stages. This data also depicts that Zs-Tra2 may have other functions which is why it is present in all stages including in males albeit at a lower level than in females.

Response: The gel picture came from RT-PCR indicating the presnece of the transcript. Bar graph was obtained from RT-qPCR, which could discriminate the mRNA amounts.It might Zs-Tra2 transcript variants. We commented the possibility of Zs-Tra2 transcript variants in the discussion. However, at this time, we cannot give any evidence to explain the transcript nature in males.

Comment #2-13: Figure 4: dsCon should also be depicted in the same gel.

Response: We showed the relative amount of Zs-Tra2 by reducing its control expression levels in RT-qPCR.

Comment #2-14: While injection of dsRNA resulted in decreased expression levels of Zs-Tra2 from 24 to 72 hrs (again, the gel does not reflect such change, but only the graph does), feeding resulted in a reversion of Zs-Tra2. A proper explanation of this is required. The effects as depicted are fairly mild.

Response: We add following explanation to discussion: “Less efficiency of feeding dsRNA might be explained by less amount of dsRNA targeting mRNA compared to injecting dsRNA because the environmental RNAi, in general, the efficiency of RNAi depends on the amount of exogenous dsRNA application [44].”

Comment #2-15: Line 270-271. What % females developed male-characteristics?

Response: We do not know exact proportion because we did not assess karyotypes. Our results suggested over 30% increase of males. This information is added to the text as follows: “Compared to control, RNAi-treated flies had more males by over 30%.”

Comment #2-16: Why did the authors not try to make the normal bacterial gut flora in  Scutellata (choosing any one predominant bacteria) express the dsRNA?

Response: This is a nice comment. It is a challenge for us to handle non-E. coli system.

Round 2

Reviewer 2 Report

The authors have addressed the concerns expressed by the reviewers in the revised submission.